# $Q$-LEARNING WITH REGULARIZATION CONVERGES WITH NON-LINEAR NON-STATIONARY FEATURES

## ABSTRACT

The deep $Q$-learning architecture is a neural network composed of non-linear hidden layers that learn features of states and actions and a final linear layer that learns the $Q$-values of the features. The parameters of both components can possibly diverge. Regularization of the updates is known to solve the divergence problem of fully linear architectures, where features are stationary and known a priori. We propose a deep $Q$-learning scheme that uses regularization of the final linear layer of architecture, updating it along a faster time-scale, and stochastic full-gradient descent updates for the non-linear features at a slower time-scale. We prove the proposed scheme converges with probability 1. Finally, we provide a bound on the error introduced by regularization of the final linear layer of the architecture.

## 1 INTRODUCTION

The $Q$-learning algorithm, introduced in the seminal paper of Watkins & Dayan (1992), is a stochastic semi-gradient descent algorithm that allows agents to learn to make sequential decisions towards long term goals by learning the optimal state-action value function of a given problem. The relevance of $Q$-learning in reinforcement learning (RL) cannot be overstated, as $Q$-learning with deep neural networks sustains the biggest breakthrough the field has seen (Mnih et al., 2015).

We can cast a deep $Q$-learning architecture as a neural network that combines a non-linear component of hidden layers, learning features of the input, and a final linear component that learns the $Q$-values of the learned features, as depicted in Figure 1. Despite its merits, there is no guarantee that $Q$-learning with function approximation architectures, even linear ones, converges to the desired solution. In fact, divergence happens in well known examples where the parameters of the approximator do not approach any solution, either oscillating within a window (Boyan & Moore, 1995; Gordon, 2001) or growing without bound (Tsitsiklis & Van Roy, 1996; Baird, 1995). There is also evidence for convergence to incompetent solutions (van Hasselt et al., 2018).

Recently, the works of Carvalho et al. (2020), Zhang et al. (2021) and Lim et al. (2022) provided insights on the role of regularization of the parameters and of the $Q$-values for stabilizing $Q$-learning with linear function approximation and obtaining a provably convergent scheme. Under the light of the architecture in Figure 1, their setting is one in which the features are stationary and known a priori and only the final component is learned. In this work, we investigate whether a regularized version $Q$-learning with linear function approximation schemes converge while features are non-stationary.

In Section 3, as a first result, we assume that the features are updated along a slower time-scale than the final layer and that they converge, and prove that the final layer converges. Our setting and proof are based on two time-scale stochastic approximation ideas. We also bound the distance between the optimal $Q$-function, generally outside the span of the features, and the regularized solution.

Then, in Section 4, we investigate how we can learn the non-linear features along the slow time-scale with provable convergence guarantees, thus verifying the assumption of the first result. We propose three learning schemes that perform stochastic full-gradient descent on well defined loss functions and are able to use a recent result from Mertikopoulos et al. (2020) to establish their convergence.

Putting our two results together, we obtain the first convergence result for stochastic semi-gradient $Q$-learning schemes with non-linear function approximation. Our scheme is two time-scale, where the final layer of a neural network is updated faster and learns regularized $Q$-values of non-linear features that are updated slower and with stochastic full-gradient descent updates.

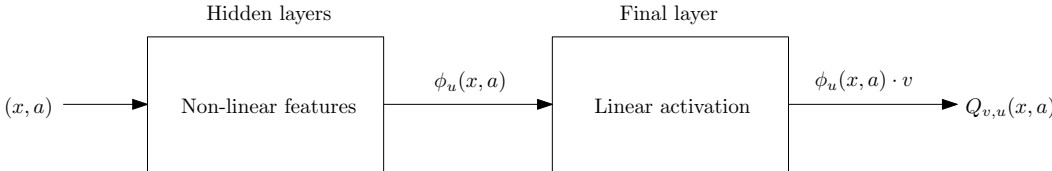

Figure 1: A general deep $Q$-learning neural network architecture. The state-action pair inputs $(x, a)$ are fed to non-linear hidden layers that output features of the input $\phi_u(x, a)$. Then a linear activation layer parameterized by $v$ outputs the $Q$-values of $(x, a)$ using the features. Usually, the architecture is learned through performing stochastic semi-gradient descent updates on the Bellman error.

## 2 BACKGROUND

A Markov decision process $\mathcal{M}$ is a tuple $(\mathcal{X}, \mathcal{A}, \mathcal{P}, \mathcal{R})$, where $\mathcal{X}$ is a finite set of states, $\mathcal{A}$ is a finite set of actions, $\mathcal{P}$ is a set of transition probability distributions $P(x, a) \in \Delta(\mathcal{X})$[1] and $\mathcal{R}$ is a set of bounded real-valued random variables, $R(x, a) \in [-r_{max}, r_{max}]$, called the reward.

An agent interacts, discretely, with an environment described as a Markov decision process by observing the random state of the process $X_t$, performing a random action $A_t$ and receiving a random reward $R_t \sim R(X_t, A_t)$. The state of the process changes to $X_{t+1}$ and the interaction repeats. The way the agent selects actions once it observes a state is prescribed by a policy $\pi$ that, for each state, is a probability distribution $\pi(x) \in \Delta(\mathcal{A})$.

For a given policy $\pi$, we measure the value of performing some action at some state through the function $Q^\pi : \mathcal{X} \times \mathcal{A} \to \mathbb{R}$. The $Q$-function, given a state-action pair $(x, a)$, gives the expected sum of rewards the agent receives throughout its interaction with the environment, after performing action $a$ in state $x$, then continuing choosing actions according to $\pi$, and considering a discount factor $\gamma \in [0, 1)$.

The Markov decision problem is the one of finding a policy $\pi^*$ such that, for every state, maximizes the value of the best action. Such policy is known to exist (Puterman, 2005, Section 6.2). It may, however, not be unique. While an optimal policy $\pi^*$ is not necessarily unique, the optimal value $Q^*$ is and verifies the fixed-point equation of the Bellman operator

$$Q^*(x, a) = \mathbb{E}\Big[R(x, a) + \gamma \max_{a' \in \mathcal{A}} Q^*(X', a')\Big], \tag{1}$$

where $X' \sim P(x, a)$ and the expectation is taken with respect to $X'$ and $R(x, a)$. Additionally, from $Q^*$ we can obtain an optimal policy $\pi^*$ by greedily choosing, for each state, the action with highest $Q$-value. Consequently, we can solve the Markov decision problem by solving the fixed-point equation above.

To solve equation 1, we can define a loss function $h : \mathbb{R}^L \to \mathbb{R}$ such that

$$h(w) = \frac{1}{2}\mathbb{E}\left[\Big(R(X, A) + \gamma \max_{a' \in \mathcal{A}} Q_w(X', a') - Q_w(X, A)\Big)^2\right], \tag{2}$$

where $Q_w : \mathcal{X} \times \mathcal{A} \to \mathbb{R}$ is a function approximator, for instance a neural network, and $w \in \mathbb{R}^L$ its parameters. Additionally, the expectation is taken with respect to a distribution over state, action, next-state and reward transitions $(X, A, X', R(X, A))$.

If we assume the off-policy target $R(x, a) + \gamma \max_{a' \in \mathcal{A}} Q_w(x', a')$ of equation 2 is fixed.[2], we obtain a stochastic semi-gradient descent scheme to minimize $h$. The resulting algorithm is called $Q$-learning with function approximation and takes the form

$$w_{t+1} = w_t + \alpha_t \big(r_t + \gamma \max_{a' \in \mathcal{A}} Q_w(x'_t, a') - Q_w(x_t, a_t)\big) \nabla_w Q_w(x_t, a_t)$$

where state-action samples $(x_t, a_t) \sim \mu$, the data distribution $\mu \in \Delta(\mathcal{X} \times \mathcal{A})$, next-states $x'_t \sim P(x_t, a_t)$, rewards $r_t \sim R(x_t, a_t)$ and $\alpha_t \in \mathbb{R}^+$ is a learning rate.

---

[1] We use $\Delta(\mathcal{B})$ to denote the set of probability distributions over a set $\mathcal{B}$.

[2] The technique is usually referred to as bootstrapping.

## 3 LEARNING WITH NON-STATIONARY FEATURES

In our work, we consider parameterized functions as depicted in Figure 1. We observe that a state-action pair $(x, a)$ is the input, and is processed by non-linear features parameterized by $u$, $\phi_u$. Then, $\phi_u(x, a)$ is passed on to a linear layer parameterized by $v$. The output is $Q_{v,u}(x, a)$. In our architecture, we use $Q_{v,u} : \mathcal{X} \times \mathcal{A} \to \mathbb{R}$ such that $Q_{v,u}(x, a) = \phi_u(x, a) \cdot \text{Proj}_\rho(v)$, where $\text{Proj}_\rho : \mathbb{R}^K \to B_\rho$ maps $v$ to a ball of radius $\rho$ in $\mathbb{R}^K$ that can be arbitrarily large. $\text{Proj}_\rho$ ensures boundedness of the $Q$-values without requiring boundedness of the parameters. We refer to the parameters of the final linear layer, $v \in \mathbb{R}^K$, as the final parameters, to the parameters of the non-linear hidden layers, $u \in \mathbb{R}^D$, as the hidden parameters and to $\phi_u : \mathcal{X} \times \mathcal{A} \to \mathbb{R}^K$ as the features.

In this section, we make the argument that learning the final parameters $v$ at a faster time-scale than the hidden parameters $u$ allows us to decouple the convergence analysis of the two and consequently establish that a regularized version of $Q$-learning with convergence guarantees for the stationary features case, will also converge if the features are non-stationary but convergent. Specifically, following Borkar (2008, Chapter 6), assuming the features change slower than the final layer allows us to treat the former as being quasi-static from the point of the view of the latter, even though both learning processes evolve simultaneously. We note that, in our analysis, we do not require, necessarily, that the features learn through the same supervision signal as the final layer.

We define the $Q$-learning scheme of the final linear layer with the addition of ridge regularization of the parameters, merging ideas from Lim et al. (2022) and Zhang et al. (2021).

**Definition 1.** In $Q$-learning with regularization, the final parameters are updated according to

$$v_{t+1} = v_t + \alpha_t \big( r_t + \gamma \max_{a' \in \mathcal{A}} Q_{v_t,u_t}(x_t', a') - \xi Q_{v_t,u_t}(x_t, a_t) \big) \phi_{u_t}(x_t, a_t) - \alpha_t \epsilon v_t,$$

where $\xi, \epsilon > 0$ are regularization hyper-parameters and the positive learning rates in $\{\alpha_t\}_{t \geq 0}$ are such that $\sum_{t=0}^{\infty} \alpha_t = \infty$ and $\sum_{t=0}^{\infty} \alpha_t^2 < \infty$. We observe that if $\xi = 1$, $\epsilon \to 0$, and $\rho \to \infty$, we recover the original $Q$-learning with linear function approximation algorithm, which can diverge.

For our first result, let us assume the features are updated much slower than the final layer through a stochastic approximation scheme with well-behaved noise. Formally, we assume the following.

**Assumption 1.** The hidden parameters $u$ are updated according to

$$u_{t+1} = u_t + \beta_t \big( g(v_t, u_t) + N_{t+1} \big),$$

where the vector field $g : \mathbb{R}^{K+D} \to \mathbb{R}^D$ is Lipschitz-continuous; the sequence of random vectors $\{N_t\}_{t \geq 0}$ has zero mean and finite variance; the learning rates in $\{\beta_t\}_{t \geq 0}$ are such that $\sum_{t=0}^{\infty} \beta_t = \infty, \sum_{t=0}^{\infty} \beta_t^2 < \infty$ and $\beta_t / \alpha_t \to 0$. We finally assume the sequence $\{u_t\}_{t \geq 0}$ converges, i.e., that $u_t \to u^*$ for some $u^* \in \mathbb{R}^D$.

In Sec.4, we propose three feature learning schemes that satisfy Assumption 1, i.e., specific choices of $g$ and $N$ that are provably convergent. For now, we establish that $Q$-learning with regularization converges with non-stationary convergent features.

**Theorem 1.** *Suppose that Assumption 1 holds and moreover*

(i) *For all $t \geq 0$, the distribution $\mu \in \Delta(\mathcal{X} \times \mathcal{A})$ and is such that $(X_t, A_t) \sim \mu$ and are i.i.d.;*

(ii) *The architecture $\phi : \mathbb{R}^D \times \mathcal{X} \times \mathcal{A} \to \mathbb{R}^K$ and is Lipschitz-continuous on the first argument;*

(iii) *For all $u \in \mathbb{R}^D$, the features $\phi_u$ are such that, for all $(x, a) \in \mathcal{X} \times \mathcal{A}$, $\|\phi_u(x, a)\| \leq 1$;*

(iv) *For all $u \in \mathbb{R}^D$, the features $\phi_u$ and the distribution $\mu$ are such that the $K \times K$ matrix $\Sigma_u := \mathbb{E}\big[\phi_u(X, A)\phi_u^T(X, A)\big]$ is positive-definite and its minimum eigenvalue is $\sigma$.*

*Additionally suppose that the regularization parameter $\xi > 1$ is large enough, specifically that $\xi > \frac{\gamma}{\sigma}$; that $\epsilon > 0$ is small enough, specifically that $\epsilon < \xi\sigma - \gamma$; and that the radius of the ball $B_\rho$, $\rho > 0$, is also large enough, specifically that $\rho > \frac{r_{max}}{\xi\sigma - \gamma - \epsilon}$.*

*Then, it holds that $v_t \to v^*(u^*)$ w.p.1, with $v^* : \mathbb{R}^D \to \mathbb{R}^K$ such that*

$$v^*(u) = \frac{1}{\xi} \Sigma_u^{-1} \mathbb{E}\Big[\big(R(X, A) + \gamma \max_{a' \in \mathcal{A}} Q_{v^*(u),u}(X', a')\big)\phi_u(X, A)\Big] - \frac{\epsilon}{\xi} \Sigma_u^{-1} v^*(u).$$

Before we present the proof, we discuss Assumptions (i) to (iv) of Theorem 1.

While not necessary, Assumption (i) facilitates the formal analysis of the stochastic processes, as well as its exposition in the document. Assumption (ii) allows us to use theoretical results on the convergence of stochastic approximation processes, such as the ones of Borkar (2008, Chapter 6; Theorem 2) and Mertikopoulos et al. (2020, Theorem 2). Assumption (iii) is important to the formal analysis of the limiting ordinary differential equations, as well as to be assured of technical requirements, such as Lipschitz continuity of the update. Specifically, since the $Q$-learning update considered features products between $v$ and $u$ dependent quantities, the assumption ensures the expected update is Lipschitz-continuous. Finally, Assumption (iv) is used to guarantee existence of solution to the limiting o.d.e., as well as to characterize such solution.

*Proof.* We present an outline of the proof, referring to the supplementary material for proofs of auxiliary lemmas.

The $Q$-learning algorithm presented is a two time-scale stochastic approximation algorithm where the fast component takes the form

$$v_{t+1} = v_t + \alpha_t \big( f(v_t, u_t) + M_{t+1} \big), \tag{3}$$

with $f : \mathbb{R}^{K+D} \to \mathbb{R}^K$ the expected update

$$f(v, u) = \mathbb{E}\Big[ \big( R(X, A) + \gamma \max_{a' \in \mathcal{A}} Q_{v,u}(X', a') - \xi Q_{v,u}(X, A) \big) \phi_u(X, A) \Big] - \epsilon v$$

and $M_t \in \mathbb{R}^K$ its noise.

Borkar (2008, Chapter 6; Theorem 2) provides conditions under which the stochastic process above converges. The conditions include that $f$ is Lipschitz and $M_{t+1}$ is a martingale-difference sequence, which we show through Lemmas 1 and 2, respectively. In addition, Lemma 3 establishes that, for each $u \in \mathbb{R}^D$, the ordinary differential equation (o.d.e.)

$$\dot{v}_t = f(v_t, u)$$

has a unique globally asymptotically stable equilibrium $v^*(u)$, using a Lyapunov argument. Since we show using Lemma 4 that, additionally, the iterates remain bounded, we conclude through Lemma 5 that they converge to the equilibrium w.p.1. □

We finish this section by providing a bound on the solution obtained by the $Q$-learning scheme considered and the optimal solution.

Let us denote the optimal $Q$-function by $Q^*$, which exists and is unique and is generally outside the linear space generated by the features $\phi_{u^*}$. Let us define the orthogonal projection of $Q^*$ into such linear space as the operator $\Phi_{u^*}$ such that $(\Phi_{u^*}Q)(x, a) = \phi_{u^*}^T(x, a)\Sigma_u^{-1}\mathbb{E}\big[\phi(x, a)Q(x, a)\big]$. We can think of $\Phi_{u^*}Q^*$ as the best linear approximator of $Q^*$. Unfortunately, such approximator is, in general, not reachable through $Q$-learning.

Using $w^*$ to jointly denote the parameters $v^*(u^*), u^*$, we have the following result for the solution $Q_{w^*}$ otained by the regularized $Q$-learning scheme.

**Theorem 2.** *Under Assumptions 1 and (i) to (iv) of Theorem 1, we have the following error bound on $Q_{w^*}$:*

$$\|Q^* - Q_{w^*}\|_\infty \leq \frac{\xi\sigma}{\xi\sigma - \gamma}\|Q^* - \Phi_{u^*}Q^*\|_\infty + \frac{r_{max}(\xi - 1)}{(1 - \gamma)(\xi\sigma - \gamma)} + f_\epsilon, \tag{4}$$

*where $f_\epsilon = \frac{\epsilon r_{max}}{\xi\sigma(\sigma - \gamma - 1)} \cdot \frac{\xi\sigma}{\xi\sigma - \gamma}$.*

In equation 4, we observe the bound depends on the regularization, through the hyper-parameters $\xi$ and $\epsilon$, and on the features, through $\sigma$ and $u^*$. We can make $\epsilon \to 0$ and make $f_\epsilon$ arbitrarily small. As for the second term, it disappears if $\xi = 1$ and the error then depends only on the best possible solution for the given features, $\Phi_{u^*}Q^*$. However, if $\xi = 1$, the $Q$-learning scheme may diverge.

*Proof.* We have that

$$\|Q^* - Q_{w^*}\|_\infty \leq \|Q^* - \Phi_{u^*}Q^*\|_\infty + \|\Phi_{u^*}Q^* - Q_{w^*}\|_\infty.$$

Let us consider the second term on the right-hand side. We know that $Q^* = HQ^*$, using $H$ to denote the Bellman operator, and that $Q_{w^*} = \frac{1}{\xi}\Phi_{u^*}HQ_{w^*} + \frac{\epsilon}{\xi}\Sigma_{u^*}^{-1}v^*(u^*)$ from the characterization of the limit solution in Theorem 1. Then, we have that

$$\|\Phi_u Q^* - Q_{w^*}\|_\infty \leq \|\Phi_{u^*}Q^* - \frac{1}{\xi}\Phi_{u^*}Q^*\|_\infty + \frac{1}{\xi}\|\Phi_{u^*}HQ^* - \Phi_{u^*}HQ_{w^*}\|_\infty + \frac{\epsilon}{\xi\sigma}\|v^*(u^*)\|. \tag{5}$$

by means of the Cauchy-Schwarz and Jensen inequalities. For the first term on the right hand side of equation 5, we can establish that

$$\|\Phi_{u^*}Q^* - \frac{1}{\xi}\Phi_{u^*}Q^*\|_\infty \leq (1 - \frac{1}{\xi})\|\Phi_{u^*}Q^*\|_\infty \leq \frac{r_{\max}(\xi - 1)}{\xi\sigma(1 - \gamma)}.$$

For the second term on the right hand side of equation 5, we have that

$$\frac{1}{\xi}\|\Phi_{u^*}HQ^* - \Phi_{u^*}HQ_{w^*}\|_\infty \leq \frac{\gamma}{\xi\sigma}\|Q^* - Q_{w^*}\|_\infty.$$

Finally, for the third term on the right hand side of equation 5, we start by noting that

$$\|v^*(u^*)\| \leq \frac{1}{\sigma}\big(r_{\max} + \gamma\|v^*(u^*)\|_\infty\big) - \frac{\epsilon}{\xi\sigma}\|v^*(u^*)\|_\infty.$$

Equivalently, for the third term on the right hand side of equation 5 we have

$$\frac{\epsilon}{\xi\sigma}\|v^*(u^*)\| \leq \frac{\epsilon r_{\max}}{\xi\sigma(\sigma - \gamma) - \epsilon\sigma}.$$

Putting everything together, we conclude the result. $\qquad\square$

### 3.1 EXPERIMENTAL RESULTS

We illustrate our proposed learning architectures under three examples with converging features. In all of them, the original $Q$-learning diverges while the proposed architecture does not.

### LINEAR $v \to 2v$ EXAMPLE

In the $v \to 2v$ example of Tsitsiklis & Van Roy (1996) there are two states and a single action. The first state always transitions to the second; the second state always transitions to itself. All rewards are 0 and, consequently, so are the $Q$-values. We consider the features $\phi_u(x) = \psi(x) + u$, where $\psi(x)$ is 1 for the first state and 2 for the second state and $u \in \mathbb{R}$. We consider $u_t = \frac{(-1)^t}{t} \to 0$. We divide the features by 2 in order to respect Assumption (iii). The desirable behavior of $Q$-learning would be $v \to 0$. Figure 2a shows the results. We can see that when $\xi = 1$ the parameter $v$ diverges. As $\xi$ increases, learning is more stable. When $\xi = 2$, $v$ converges to the desired solution $v = 0$.

### STAR EXAMPLE

The star example of Baird (1995), slightly modified by Sutton & Barto (2018), has seven states and two actions. One of the actions transitions to any of the first six states uniformly, the other action transitions to the seventh state. All rewards are 0 and so are the $Q$-values. The behavioral policy chooses the first action with probability $\frac{6}{7}$ and the second action with probability $\frac{1}{7}$. Therefore, the next state distribution is uniform. The target policy, however, always chooses the second action. For the first six states, the state-features features are $\phi_u(x) = \psi(x) + u \in \mathbb{R}^8$ where $\psi(x)$ are 2 in the $x$-th component, 1 in the eight component and 0 otherwise. For the seventh state, the features are 1 in the seventh component and 2 in the eight component. We consider again converging hidden parameters $u_t = (-1)^t(\frac{1}{t}, \frac{1}{t}, \frac{1}{t}, \frac{1}{t}, \frac{1}{t}, \frac{1}{t}, \frac{1}{t}, \frac{1}{t}) \to 0$. We divide the features by $\sqrt{5}$ in order to respect Assumption (iii). Figure2b shows the results obtained. When $\xi = 1$, the parameters $v$ grow. However, we see that as $\xi$ increases, the final parameters $v$ do not grow, as desired.

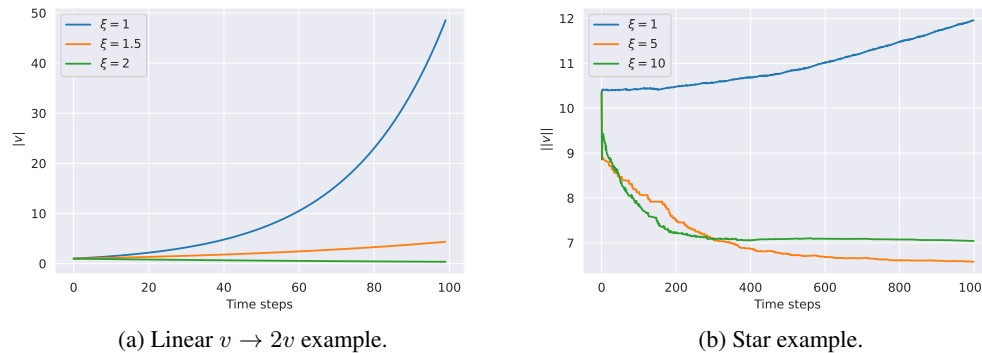

(a) Linear $v \to 2v$ example.

(b) Star example.

Figure 2: Experimental results on the linear $v \to 2v$ and star problems under non-stationary convergent features for different values of regularization parameters $\xi$ and fixed $\epsilon = 10^{-8}$. As the regularization parameter $\xi$ increases, $Q$-learning updates stabilize and the parameters $v$ converge.

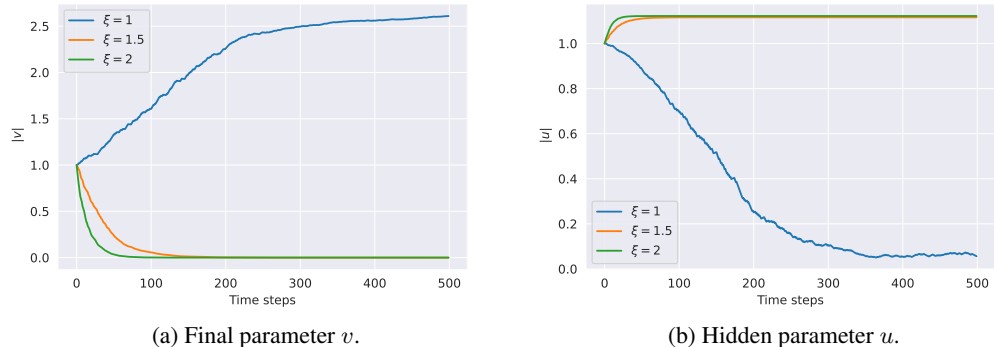

(a) Final parameter $v$.

(b) Hidden parameter $u$.

Figure 3: Experimental results on the non-linear $v \to 2v$ problem for different values of regularization parameters $\xi$ and fixed $\epsilon = 10^{-8}$. As the regularization parameter $\xi$ increases, $Q$-learning updates stabilize and the final parameters $v$ converge. The hidden parameter $u$ converges regardless of the regularization parameter $\xi$, though the limit solutions are different for different values of $\xi$.

### NON-LINEAR $v \to 2v$ EXAMPLE

We modify the $v \to 2v$ to a non-linear learning architecture. The Markov chain remains the original from Tsitsiklis & Van Roy (1996), and so does the data distribution. We learn the linear features with sigmoid activation function $\phi_u(x) = \sigma(\psi(x)u)$ with the final linear layer parameterized by $v$ and again $\psi(x) = x$. Then, we have $Q_{v,u}(x) = \sigma(\phi(x)u)v$. $v = 0$ recovers the correct $Q$-values. In Figure 3a, we see divergence of the final parameter when $\xi = 1$ and convergence to the correct solution when $\xi = 1.5$ and $\xi = 2$. In both cases the features converge, as can be seen is Figure 3b.

## 4 LEARNING NON-LINEAR FEATURES

Theorem 1 states that the regularized $Q$-learning scheme is convergent with features that are changing over time, throughout the learning process, as long as those features converge. We now present three learning settings for the hidden layers that we can show to satisfy the assumption, i.e., three learning setting under which convergence of the features is guaranteed.

We consider a $D$-times differentiable objective function $h : \mathbb{R}^D \to \mathbb{R}$. We want to find $z^*$ such that

$$z^* = \min_{z \in \mathbb{R}^D} h(z).$$

The stochastic gradient descent scheme for solving the equation above takes the form

$$z_{t+1} = z + \beta_t\big(\nabla_z h(z_t) + Y_{t+1}\big),$$

where the random variables $Y_t$ have zero-mean and bounded variance. We have the following result from Mertikopoulos et al. (2020).

**Theorem 3.** *Suppose that the function $h$ is Lipschitz-continuous, Lipschitz-smooth, coercive and not asymptotically flat. Then, we have that the set of critical points $Z^* := \{z : \nabla_z h(z) = 0\}$ is non-empty. Further suppose that the random variables $Y_t$ have zero-mean and finite variance. Then,*

$$z_t \to Z_\infty^* \quad w.p.1,$$

*where $Z_\infty^* \subseteq Z^*$ is a bounded connected component over which $h$ is constant.*

Theorem 3 provides general conditions under which the parameters of a stochastic approximation scheme that is, particularly, stochastic gradient descent of a loss function $h$, converge to a bounded region with constant value. While $Q$-learning is not true stochastic gradient descent and divergence of the parameters is known to happen, it is possible to learn the parameters of the features through stochastic full-gradient descent and obtain convergence guarantees. In the sequel, we propose three such feature learning schemes. One of the proposed schemes is based on an unupervised learning update; another is based on a semi-supervised learning update; the final is based on a reinforcement learning update. In light of Theorem 3, all the proposed feature learning schemes are in accordance with Assumption 1 and are thus guaranteed to converge. We note, however, that in order to guarantee boundedness of the features, we should post-process them with a sigmoid final layer, $\sigma : \mathbb{R} \to [0,1]$ such that $\sigma(x) = \frac{1}{1+e^{-x}}$, thus respecting Assumption (iii) of Theorem 1.

We define the generalized Huber loss $H : \mathbb{R}^L \to \mathbb{R}$ such that $H(l) = \min_{p \in \{1,2\}} \frac{1}{p} \|l\|_p^p$ is a robust loss function in the conditions demanded by the theorem. The Huber loss considered is thus the 2-norm if its input is close to the origin and the 1-norm otherwise. Additionally, we remark that the finite linear combination and the finite composition of Lipschitz-continuous and Lipschitz-smooth functions is also Lipschitz-continuous and Lipschitz-smooth. Finally, we note that $\nabla H(l) = sign(l)$ if $\arg\min_{p \in \{1,2\}} \frac{1}{p} \|l\|_p^p = 1$ and $\nabla H(l) = l$ if $\arg\min_{p \in \{1,2\}} \frac{1}{p} \|l\|_p^p = 2$.

## 4.1 UNSUPERVISED LEARNING

We can learn a linear map that reduces the input space, linearly, through principal component analysis. In the non-linear case, we can instantiate such learning using auto-encoder (Liou et al., 2014) or variational auto-encoder (Kingma & Welling, 2014) architectures. Finally, constrastive learning has also been used in feature extraction (Laskin et al., 2020). All these methods have no task information but can still be powerful if dimensionality is an issue or we want to transfer learning across tasks or even domains (Higgins et al., 2017). We focus here on the definition of a loss function over which the stochastic gradient descent scheme is guaranteed to converge: the auto-encoder. Formally, the auto-encoder performs stochastic gradient descent over the loss function

$$h(u,s) = \mathbb{E}\left[ H\left( \kappa_s\big(\phi_u\big(\psi(X,A)\big)\big) - \psi(X,A) \right) \right],$$

where $\psi(x,a)$ is an euclidean representation of $(x,a)$ in $\mathbb{R}^P$. In the auto-encoder, an encoder $\phi_u : \mathbb{R}^P \to \mathbb{R}^K$, $u \in \mathbb{R}^D$ learns to map the features into a latent space and a decoder $\kappa_s : \mathbb{R}^K \to \mathbb{R}^P$ learns to reconstruct the original input. The features $\phi_u$ can then be normalized and inputted to the final layer of our regularized $Q$-learning scheme. The auto-encoder has been applied successfully in reinforcement learning tasks (Lange et al., 2012).

In practice the stochastic gradient updates are as follows.

$$u_{t+1} = u_t - \beta_t \nabla_u \phi_{u_t}\big(\psi(x,a)\big) \nabla \kappa_{s_t}\Big(\phi_{u_t}\big(\psi(x,a)\big)\Big) \nabla H\Big( \kappa_s\big(\phi_u\big(\psi(x,a)\big)\big) - \psi(x,a) \Big)$$

$$s_{t+1} = s_t - \beta_t \nabla_s \kappa_{s_t}\Big(\phi_u\big(\psi(x,a)\big)\Big) \nabla \phi_{u_t}\big(\psi(x,a)\big) \nabla H\Big( \kappa_s\big(\phi_u\big(\psi(X,A)\big)\big) - \psi(X,A) \Big).$$

The process of learning the final linear layer parameterized by $v$ evolves concurrently as described in Section 3. As long as $\beta_t$ is $o(\alpha_t)$, convergence of both sets of parameters happens with probability 1 according to Theorem 3.

### 4.2 SEMI-SUPERVISED LEARNING

We can also approach the feature learning problem in a semi-supervised way grounded in MDP theory (Ferns et al., 2004). Specifically, instead of only learning to put together inputs that are close to each other in the original space, we can learn to put them together if they are close to each other in the Markov decision process. Bissimulation metrics (Ferns et al., 2011) give us a way to perform such learning, by considering that state-action pairs are similar if they produce similar rewards and lead to similar states. We can thus define the loss function

$$h(u) = \mathbb{E}\left[ H\Big( H\big(\phi_u(X, A) - \phi_u(\tilde{X}, \tilde{A})\big) - H(R - \tilde{R}) - \gamma H\big(\phi_u(X', \cdot) - \phi_u(\tilde{X}', \cdot)\big)\Big)\right]$$

and perform again stochastic gradient descent. The technique has also provided positive results in practice, specifically when compared to unsupervised learning (Zhang et al., 2020).

In this semi-supervised setting, the stochastic gradient updates take the form

$$u_{t+1} = u_t - \beta_t\Big(\nabla_u\big(\phi_{u_t}(x_t, a_t) - \phi_{u_t}(\tilde{x}_t, \tilde{a}_t)\big)\nabla H\big(\phi_{u_t}(x_t, a_t) - \phi_{u_t}(\tilde{x}_t, \tilde{a}_t)\big) -$$

$$- \gamma \nabla_u\big(\phi_{u_t}(x_t, a_t) - \phi_{u_t}(\tilde{x}_t, \tilde{a}_t)\big)\nabla H\big(\phi_{u_t}(x_t, \cdot) - \phi_{u_t}(\tilde{x}_t, \cdot)\big)\Big)$$

$$\nabla H\Big( H\big(\phi_{u_t}(x_t, a_t) - \phi_{u_t}(\tilde{x}_t, \tilde{a}_t)\big) - H(r - \tilde{r}_t) - H\big(\phi_{u_t}(x_t, \cdot) - \phi_{u_t}(\tilde{x}_t, \cdot)\big)\Big),$$

where $(\tilde{x}_t, \tilde{a}_t, \tilde{x}'_t, \tilde{r}_t)$ are sampled independent and identically distributed. Again, the learning scheme used to update the hidden parameters $u$ can happen alongside learning of the final linear layer parameterized by $v$ and convergence is guaranteed by Theorem 3.

### 4.3 REINFORCEMENT LEARNING

$Q$-learning assumes bootstrapped targets and thus makes only stochastic semi-gradient descent over the loss function. In practice, that choice may produce good results but is often unstable. We propose that we could learn the features through full stochastic gradient descent and learn the final layer through the usual regularized stochastic semi-gradient descent scheme. The loss function considered is

$$h(u, v') = \mathbb{E}\Big[ H\big(R + \gamma \max_{a' \in \mathcal{A}} \phi_u(X', a') \cdot v' - \phi_u(X, A) \cdot v'\big)\Big].$$

Recent work from Avrachenkov et al. (2021) proves that, often, the stochastic full gradient descent over the loss function is able to perform competitively with the stochastic semi-gradient scheme. The updates take the form

$$u_{t+1} = u_t + \beta_t \nabla H\big(r_t + \gamma \max_{a' \in \mathcal{A}} \phi_{u_t}(x'_t, a') \cdot v'_t - \phi_{u_t}(x_t, a_t) \cdot v'_t\big)$$

$$\big(\nabla_u \phi_{u_t}(x_t, a_t)v'_t - \gamma \nabla_u \max_{a' \in \mathcal{A}} \phi_{u_t}(x'_t, a')v'_t\big)$$

$$v'_{t+1} = v'_t + \beta_t \nabla H\big(r_t + \gamma \max_{a' \in \mathcal{A}} \phi_{u_t}(x'_t, a') \cdot z_t - \phi_{u_t}(x_t, a_t) \cdot v'_t\big)$$

$$\big(\phi_{u_t}(x_t, a_t) - \gamma \nabla_v \max_{a' \in \mathcal{A}} \phi_{u_t}(x'_t, a')v'_t\big).$$

For the gradient of the $\max$ operator, we may consider a smooth approximation parameterized by $\alpha$, $\max_\alpha$, such that $\max_\alpha(v_1, \ldots, v_n) = \frac{\sum_{i=1}^n v_i e^{\alpha v_i}}{\sum_{i=1}^n e^{\alpha v_i}}$. As $\alpha \to \infty$, $\max_\alpha \to \max$.

The hidden parameters $u$ that are being updated can, at the same time, be used to learn the final parameters $v$ using the regular semi-gradient $Q$-learning update. In the regularized version of $Q$-learning presented in Section 3, convergence happens (Theorem 3).

In practice, our proposed approach is halfway between the full-gradient and semi-gradient schemes for reinforcement learning and is able to capture the stability of full-gradient schemes and optimality of semi-gradient schemes. The additional computational and training costs are minimal. More specifically, one additional linear layer of parameters $z \in \mathbb{R}^K$ is required.

## 5 RELATED WORK

There were initial efforts to provide stable $Q$-learning methods in the presence of function approximation (Singh et al., 1994; Ormoneit & Sen, 2002; Szepesvári & Smart, 2004; Melo et al., 2008; Maei et al., 2010). While meritous and insightful, the referred works were not only restricted to the linear function approximation case but also assumed the data was particularly well-aligned with specific distributions (Melo et al., 2008).

More recently, Q-learning was attributed finite-time error bounds when certain fixed behavior policies are used Chen et al. (2019). Such policies are scarce or may not even exist as the number of features grows. Finite-time error bounds for Adaptive Dynamic Programming methods Bertsekas & Tsitsiklis (1995), including Fitted Q-iteration (FQI) Ernst et al. (2005), assume not only the realizability of the optimal Q-function but also closedness under Bellman update Szepesvári & Munos (2005). Such conditions have been discussed in a recent work Chen & Jiang (2019).

The problem of divergence of $Q$-learning with function approximation was revived after a significant empirical success story of the use of $Q$-learning with deep neural networks (Mnih et al., 2015), where the function approximation setting is non-linear and the features are non-stationary. One of the components of the renowned deep $Q$-network (DQN) is a target network that mitigates the negative impact of bootstrapping, i.e., of the stochastic semi-gradient update.

The works of Carvalho et al. (2020); Zhang et al. (2021); Chen et al. (2022) provided theoretical insights and convergence guarantees for $Q$-learning with the target network. There is also work pointing out that regularization of the $Q$-values or the parameters themselves can stabilize $Q$-learning, resulting in a convergent algorithm (Zhang et al., 2021; Carvalho et al., 2020; Lim et al., 2022). All these works are applicable to the linear function approximation case, merely. In the case of linear function approximation, the features are assumed to be stationary and known prior to learning. The practical applications of reinforcement learning, however, are moving in the opposite direction, where the features are also learned. Additionally, such learning setting is, typically, non-linear. Consequently, the gap between theory and practice remains significant.

For the case of non-linear function approximation, a recent work suggests a loss function that is decreasing over time, assuming the neural network converges to the targets from a target network at each step (Wang & Ueda, 2021). Finally, Xu & Gu (2020) provide a finite-time result for $Q$-learning with over-parameterized neural networks. While being an interesting result, as the size of the network grows to infinity the learning architecture also grows closer to a tabular representation.

## 6 CONCLUSION

In this work, we provided the first convergence result for a $Q$-learning scheme with non-linear non-stationary features without the use of a target network. In our scheme, the final layer of a network is updated faster than the hidden layers. We show that if the features converge, the final layer also converges. We complement our theoretical analysis with experiments showcasing our result. Finally, we propose three schemes that result in guaranteed convergence of the features.

In the future, it would be relevant to compare experimentally the three learning schemes for the features considered, specifically unsupervised, semi-supervised and full-gradient reinforcement learning. Additionally, it would be important to characterize the solutions obtained by each scheme.

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
