# OpenReview forum: "$Q$-learning with regularization converges with non-linear non-stationary features"
_ICLR.cc/2023/Conference — Submitted to ICLR 2023_

### Official Review · Reviewer_J62D · 2022-10-23

**Confidence:** 5
**Correctness:** 3
**Technical Novelty And Significance:** 1
**Empirical Novelty And Significance:** 1
**Recommendation:** 3

**Clarity, Quality, Novelty And Reproducibility:**

The paper is well written, but the results are not novel and seem to be a combination of several existing papers (Zhang et al. 2021), (Lim et al. 2022), (Mertikopoulos et al. 2020), and (Borkar 2008).

**Strength And Weaknesses:**

While the writing and the exposition are clear, there are many limitations of this work, as elaborated below.

(1) Compared to using linear function approximation, the two-layer neural network approximation enables the agent to also update the features. Therefore, one should expect the function approximation error to be smaller than simply using linear function approximation. In my point of view, this is the main advantage of using two-layer neural network, and is the motivation of this work. However, in view of Theorem 2, this advantage is not theoretically clear. In particular, it is not clear if the function approximation error $||Q^*-\Phi_{u^*}Q^*||$ is smaller compared to using linear function approximation where the features are fixed.

(2) This paper follows the regularization framework proposed in (Zhang et al. 2021) and (Lim et al. 2022). However, it was discussed in a recent work (Chen et al. 2022) Appendix E that the regularization technique proposed in (Zhang et al. 2021) is to stabilize Q-learning by changing the problem discount factor to a smaller one. This is the reason that their algorithm does not converge to $Q^*$ even in the tabular setting. Similar issue is present in (Lim et al. 2022) too. Since this paper follows that regularization framework, it has the same issue. To see this, the second and the third term on the right-hand side of Theorem 2 Eq. (4) do not vanish even in the tabular setting, suggesting that the MDP problem is no longer the original one. While this is a limitation in previous papers, since this paper follows the same framework, the validity of the algorithm design is unclear.

(3) As a follow-up concern to (2), since introducing the regularization parameter is equivalent to reducing the MDP discount factor, and it is well-known that small discount factor leads to convergence of Q-learning with linear function approximation, the numerical simulations are not surprising. Also, since DQN (Mnih et al., 2015) is known to be successful, and the authors propose a different DQN scheme, it may be a good idea to compare the empirical performance of the proposed algorithm to that of DQN (Mnih et al., 2015).

(4) Both the regularization parameters $\zeta$ and $\epsilon$, and the projection radius $\rho$ depend on the unknown parameter $\sigma$. Therefore, it is unclear how to implement this algorithm in practice.

(5) The technical contribution is incremental because (i) using regularization to overcome divergence is readily established in the literature, (ii) the convergence of the slower time-scale follows from existing literature on SGD, and (iii) the convergence of the faster time-scale follows from the ODE framework.




**Summary Of The Paper:**

This paper studies Q-learning with neural network approximation and proposes a two-layer regularized Q-learning algorithm. The algorithm can be viewed as an extension of Q-learning with linear function approximation, where an additional layer is used to learn the features. The authors show asymptotic convergence of the proposed algorithm using the ODE approach. Numerical simulations were performed on a well-known divergent example of vanilla Q-learning, on which the proposed algorithm is stable.

**Summary Of The Review:**

Since the designed algorithm (which can be viewed as an extension to (Zhang et al. 2021), (Lim et al. 2022)) suffers from the same limitations as in (Zhang et al. 2021), (Lim et al. 2022), and the technical contribution is incremental, I do not recommend this paper being accepted by ICLR.

---

> ### Author Response · Authors · 2022-11-17
> **Response to reviewer J62D**
>
> We thank the reviewer for the thoughtful comments and suggestions. We also thank the reviewer for acknowledging the quality of our exposition.
>
> We now reply to each of the reviewer's points.
>
> (1) In this work, our main motivation is to take a step from the many theoretical results on the convergence of $Q$-learning with linear function approximation towards the non-linear setting, where results are scarse. To this end, we consider non-linear layers are updated along much slower time-scales than a final linear layer and such assumption allowed us to conclude convergece in non-linear settings. Our assumption is that the non-linear hidden layers converge. In this work, we do not characterize in terms of performance the different features that can be passed on to the final linear layer. However, we agree it is an interesting direction to analyze the performance of different features, i.e., which types of updates benefit the updates and the quality of the final linear layer.
>
> (2) We thank the reviewer for directing us to the discussion in Chen et al. 2022, which is indeed relevant. In this work, our main motivation is to take a step from the many theoretical results on the convergence of $Q$-learning with linear function approximation towards the non-linear setting, where results are scarse. To this end, we consider non-linear layers are updated along much slower time-scales than a final linear layer and such assumption allowed us to conclude convergece in non-linear settings. In this case the version of $Q$-learning with linear function approximation we considered was the regularized by Lim et al. (2022) but other convergent variants of $Q$-learning with linear function approximation could be used. For example, un-regularized methods such as soft-state aggregation and interpolation based methods (Singh et al. 1994; Szepesvári and Smart 2004) can also be used to learn the final linear layer.
>
> (3) We thank the reviewer for suggesting further experimental studies comparing regularized methods and the DQN. While we agree such experiments would be interesting, in this work we focus on the extension of the convergence of $Q$-learning with linear function approximation methods to non-linear settings using the techniques of two time-scale stochastic approximation.
>
> (4) The parameter $\epsilon$ can take a very small value that is bigger than 0 without significantly affecting the quality of the solution. A good strategy for choosing the parameter $\xi$ could be to start with $\xi = 1$ and, if the parameters grow too much, increase $\xi$ and re-start the algorithm. The reason for the strategy is to trade-off convergence and performance. Finally, a sufficient condition for choosing $\rho$ is to choose it larger than $\frac{r_{\text{max}}}{1 - \gamma - \epsilon}$.
>
> (5) We respectfully disagree with the reviewer. Combining different already established results can also incur technical novelty. We believe such is not uncommon and that such is the case in our work.

---

> > ### Comment · Reviewer_J62D · 2022-11-24
> > **After Response**
> >
> > I thank the authors for their detailed response.
> >
> > (1) The (probably one and only) motivation for using two-layer neural network compared to linear function approximation is that the features can be updated to improve the approximation error. If this is not explicitly shown in the paper then what is the advantage of using two-layer neural net compared to linear function approximation?
> >
> > (2) It may be a better idea to avoid the use of regularization parameters which are equivalent to changing to the problem parameters.
> >
> > (3) and (5) I agree with the authors that for papers with theoretical nature extensive numerical simulations are not required. I also agree with authors that it is common for papers to be built on existing results. However, there should be some technical challenges in extending previous results, and novel ideas to overcome them. The technical novelties in this paper are unclear.

---

### Official Review · Reviewer_5r8F · 2022-10-24

**Confidence:** 2
**Correctness:** 3
**Technical Novelty And Significance:** 3
**Empirical Novelty And Significance:** 3
**Recommendation:** 6

**Clarity, Quality, Novelty And Reproducibility:**

This paper is clearly written and the treatment of convergence of deep Q-learning with non-stationary features is novel.

**Strength And Weaknesses:**

Strength:

1. The paper propose a novel scheme for deep Q-learning with non-stationary features and rigorously proved that it converges.

2. Three settings are considered under which the features converge.

3. Numerical results support the theoretical analysis.

Weakness:

1. A lack of (empirical) verification of the obtained bound.
2. Three settings with feature convergence are considered. How about other cases? It is suggested to add some discussions and/or limitations of the current results.

**Summary Of The Paper:**

This paper proposed a new scheme of deep Q-learning by introducing final layer regularization and then updates it along faster time-scale than that of the non-linear features. Moreover, it provably converges even when the  features are non-stationary. A bound is also derived on the error introduced by regularization.

**Summary Of The Review:**

Generally this is a good and rigorous paper on evaluating the convergence of deep Q-learning with non-stationary features. A new scheme is proposed and a following proof is given to show its convergence under some assumptions. My only concern is a lack of discussions of the generalization results for other settings apart from the three considered in current paper, i.e., a discussion of limitations and useful insights are suggested.

---

> ### Author Response · Authors · 2022-11-17
> **Response to reviewer 5r8f**
>
> We thank the reviewer for the thoughtful comments and suggestions. We also thank the reviewer for acknowledging the rigor of our theoretical analysis and the relevance of our experiments.
>
> We answer the reviewer regarding the implications of using other feature learning methods. In this work, our main motivation is to take a step from the many theoretical results on the convergence of $Q$-learning with linear function approximation towards the non-linear setting, where results are scarse. To this end, we consider non-linear layers are updated along much slower time-scales than a final linear layer and such assumption allowed us to conclude convergence in non-linear settings. Our assumption is that the non-linear hidden layers converge. In general, it is not easy to guarantee convergence of non-linear layers that are updated through the usual semi-gradient updates of reinforcement learning. Therefore, we focus on full-gradient updates for the non-linear layers. Those can be obtained either in supervised, semi-supervised and even reinforcement learning settings and we can easily establish convergence guarantees (Mertikopoulos et al. 2020). In this work, we do not characterize in terms of performance the different features that can be obtained. However, we agree it is an interesting direction to analyze the performance of different features, i.e., which types of updates benefit the updates and the quality of the final linear layer.

---

### Official Review · Reviewer_gATf · 2022-10-27

**Confidence:** 3
**Correctness:** 3
**Technical Novelty And Significance:** 2
**Empirical Novelty And Significance:** 2
**Recommendation:** 3

**Clarity, Quality, Novelty And Reproducibility:**

The paper is easy to read and the presentation is clear except for some clarification on the motivation of the algorithm design. The theoretical novelty is limited given that the results are also much weaker though the authors try to solve a harder problem than existing work for Q-learning with linear function approximation.


**Strength And Weaknesses:**

I think the problem studied here is very interesting, especially the investigation of when Q learning diverges and how to remedy it. The experimental results are a good illustration of such a goal. However, the motivation for the particular regularization is not clear. There is no sufficient discussion on how the regularization is designed and what happens when the regularization deviates from 0. The theoretical analysis in this paper seems to be sloppy and does not address the convergence of the proposed algorithm as claimed.


**Summary Of The Paper:**

This paper studies the convergence of a regularized Q learning algorithm and the features are learned from a nonlinear approximation. The features are updated much slower than the $Q$ function parameter. The authors prove that the proposed regularized Q-learning converges as long as the feature learning scheme converges and other conditions hold. Empirical results are provided to show that the proposed algorithm can converge while Q-learning diverges in some problems.


**Summary Of The Review:**

For Definition 1, it is unclear why there are two regularization parameters in the update. In particular, what is the role of $\xi$? It is clear that when $\xi=1$, the regularization disappears. But what happens with a very small or large $\xi$? Maybe the authors should motivate this regularization in the loss function before presenting it in the update rules.

What is $\rho$ in Definition 1?

The result in Theorem 1 is somewhat weak since it only holds asymptotically. In practice, the convergence could be impacted heavily by the approximation of the feature learning at each step, which is now assumed to converge in Assumption 1.

Therefore, the theoretical novelty is limited given that the results are also much weaker though the authors try to solve a harder problem than existing work for Q-learning with linear function approximation.

---

> ### Author Response · Authors · 2022-11-17
> **Response to reviewer gATf**
>
> We thank the reviewer for the thoughtful comments and suggestions. We also thank the reviewer for acknowledging the relevance of the problem we tackle.
>
> We now clarify the role of $\xi$ in $Q$-learning with regularization. The regularization parameter $\xi$ introuced by Lim et al. (2022) makes the second term of the bootstrapped target $r_t + \gamma \max_{a'}Q(x_t', a')$ smaller than $Q(x_t, a_t)$. Consequently, if $\xi$ is large enough, divergence can not happen.
>
> We also answer the reviewer in that the parameter $\rho$ appearing in Definition 1 is the same that was introduced in the first paragraph of the section, i.e., the radius of the ball considered for the final layer of the neural network architecture.
>
> Finally, we clarify that in this work, our main motivation is to take a step from the many theoretical results on the convergence of $Q$-learning with **linear** function approximation towards the **non-linear** setting, where results are scarse. To this end, we consider non-linear layers are updated along much slower time-scales than a final linear layer and such assumption allowed us to conclude convergece in non-linear settings. In this case the version of $Q$-learning with linear function approximation we considered was the regularized by Lim et al. (2022) but other convergent variants of $Q$-learning with linear function approximation could be used. For example, un-regularized methods such as soft-state aggregation and interpolation based methods (Singh et al. 1994; Szepesvári and Smart 2004) can also be used to learn the final linear layer.

---

### Decision · Program_Chairs · 2023-01-20

**Decision:**

Reject

**Justification For Why Not Higher Score:**

The reviewer seem to reach a consensus that the paper is not ready to be published in neurips. In particular, the reviewers raised following concerns.

--- "The theoretical novelty is limited given that the results are also much weaker though the authors try to solve a harder problem than existing work for Q-learning with linear function approximation."

---"A lack of (empirical) verification of the obtained bound."

---"The technical contribution is incremental because (i) using regularization to overcome divergence is readily established in the literature, (ii) the convergence of the slower time-scale follows from existing literature on SGD, and (iii) the convergence of the faster time-scale follows from the ODE framework."

---"Three settings with feature convergence are considered. How about other cases? It is suggested to add some discussions and/or limitations of the current results."

**Justification For Why Not Lower Score:**

N/A

**Metareview: Summary, Strengths And Weaknesses:**

The reviewer seem to reach a consensus that the paper is not ready to be published in neurips. In particular, the reviewers raised following concerns.

--- "The theoretical novelty is limited given that the results are also much weaker though the authors try to solve a harder problem than existing work for Q-learning with linear function approximation."

---"A lack of (empirical) verification of the obtained bound."

---"The technical contribution is incremental because (i) using regularization to overcome divergence is readily established in the literature, (ii) the convergence of the slower time-scale follows from existing literature on SGD, and (iii) the convergence of the faster time-scale follows from the ODE framework."

---"Three settings with feature convergence are considered. How about other cases? It is suggested to add some discussions and/or limitations of the current results."